# A Retrospective Evaluation to Assess Reliability of Electrophysiological Methods for Diagnosis of Hearing Loss in Infants

**DOI:** 10.3390/brainsci12070950

**Published:** 2022-07-20

**Authors:** Marco Mandalà, Luca Mazzocchin, Bryan Kevin Ward, Francesca Viberti, Ilaria Bindi, Lorenzo Salerni, Giacomo Colletti, Liliana Colletti, Vittorio Colletti

**Affiliations:** 1Otolaryngology Department, University of Siena, 53100 Siena, Italy; mandal@unisi.it (M.M.); ilaria.bindi88@gmail.com (I.B.); lorenzosalerni@gmail.com (L.S.); 2Ospedale Sacro Cuore-Don G. Calabria, Negrar, 37100 Verona, Italy; kaed@libero.it; 3Department of Otolaryngology-Head and Neck Surgery, John Hopkins Hospital, Baltimore, MD 21218, USA; bward15@jhmi.edu; 4Department of Maxillo-Facial Surgery, University of Milan, 20122 Milan, Italy; giacomo.colletti@gmail.com; 5ENT Department, Biomedical Sciences for Health, University of Milan, 20122 Milan, Italy; liliana.colletti@unimi.it; 6International Center for Performing and Teaching Auditory Brainstem Surgery in Children, 20121 Milan, Italy; vittoriocolletti@yahoo.com

**Keywords:** infants, auditory electrophysiological investigation, cochlear implant, auditory brainstem implant, reliability

## Abstract

Background: An electrophysiological investigation with auditory brainstem response (ABR), round window electrocochleography (RW-ECoG), and electrical-ABR (E-ABR) was performed in children with suspected hearing loss with the purpose of early diagnosis and treatment. The effectiveness of the electrophysiological measures as diagnostic tools was assessed in this study. Methods: In this retrospective case series with chart review, 790 children below 3 years of age with suspected profound hearing loss were tested with impedance audiometry and underwent electrophysiological investigation (ABR, RW-ECoG, and E-ABR). All implanted cases underwent pure-tone audiometry (PTA) of the non-implanted ear at least 5 years after surgery for a long-term assessment of the reliability of the protocol. Results: Two hundred and fourteen children showed bilateral severe-to-profound hearing loss. In 56 children with either ABR thresholds between 70 and 90 dB nHL or no response, RW-ECoG showed thresholds below 70 dB nHL. In the 21 infants with bilateral profound sensorineural hearing loss receiving a unilateral cochlear implant, no statistically significant differences were found in auditory thresholds in the non-implanted ear between electrophysiological measures and PTA at the last follow-up (*p* > 0.05). Eight implanted children showed residual hearing below 2000 Hz worse than 100 dB nHL and 2 children showed pantonal residual hearing worse than 100 dB nHL (*p* > 0.05). Conclusion: The audiological evaluation of infants with a comprehensive protocol is highly reliable. RW-ECoG provided a better definition of hearing thresholds, while E-ABR added useful information in cases of auditory nerve deficiency.

## 1. Introduction

Infant hearing loss is associated with multiple adverse sequelae, including poor speech and language acquisition, decreased global quality of life, and increased societal cost for the care of deaf people. Cochlear implantation (CI) is one of the most common procedures to address this problem, together with auditory brainstem implantation (ABI), with steadily improving outcomes. Yet, despite the success of cochlear implantation, several important questions remain; specifically, the early identification of which infants would benefit from cochlear implantation [1,2,3,4,5,6,7,8,9,10,11,12]. Low rates of anesthetic and perioperative complications in young children [13] in association with improved outcomes in earlier implanted children have led to dramatic increases in the number of infants being implanted at younger than 12 months of age [14,15,16,17,18,19]. The benefits of earlier implantation increase the need to accurately diagnose hearing loss in younger infants.

The primary aims of electrophysiological investigation in children with suspected hearing loss are: (1) the determination of hearing levels, and subsequently, in combination with radiological investigations, (2) the determination of candidacy for CI or ABI. Several electrophysiological tests have been proposed [20,21,22,23]. The most widely used methods for screening newborns for hearing loss are otoacoustic emissions, automated ABR and ABR during sleep, and behavioral audiograms. However, these methods may be unreliable in determining a precise hearing threshold in infants.

Since 1998, we have been developing a comprehensive electrophysiologic testing protocol. In addition to auditory brainstem responses (ABR), round window electrocochleography (RW-ECoG) and round window electrical auditory brainstem responses (RW-EABR) have been applied in a standardized manner. RW-ECoG, a near-field cochlear potential, has the advantage over ABR of better defining hearing thresholds and residual hearing levels. RW-EABR can provide important information regarding the excitability of the auditory nerve, the site of the hearing loss, and therefore the candidacy for CI or ABI.

The aim of the present study is to evaluate this auditory electrophysiological testing protocol in deaf children for the selection of candidates for auditory implantation by reviewing the long-term follow-up of the non-implanted ear.

## 2. Materials and Methods

Since 1998, we have used an electrophysiological testing protocol for determining hearing thresholds in children suspected of having profound hearing loss. This protocol was adopted in a tertiary referral center after newborn hearing screening test failure (otoacoustic emission and/or ABR during sleep). The protocol involves a series of tests including ABR, RW-ECoG, and RW-EABR performed under general anesthesia in a single recording session. The aim of the testing protocol is to exclude possible residual functional hearing, determine the site of the hearing loss, and subsequently, in association with imaging studies, select candidates for CI or ABI. If the electrophysiological data indicate that a child is a suitable candidate for CI or ABI, he or she additionally undergoes computed tomography (CT) and magnetic resonance imaging (MRI) to confirm the diagnosis and evaluate for inner ear malformations. All subjects underwent genetic, pediatric, and neuropsychiatric evaluation, in order to understand if any associated disabilities (developmental delay, cognitive disorders) or genetic factors for hearing loss were present. A flow chart for the testing protocol is shown in Figure 1. 

With our testing protocol, we identified 21 infants who were diagnosed with bilateral profound sensorineural hearing loss before 6 months of age between 2002 and 2009 and were fitted with a unilateral cochlear implant (Cochlear Nucleus Series, Cochlear Ltd., Sydney, Australia) before the age of 1 year. This group of implanted infants was used to test the efficacy of the protocol below 1 year of age through long-term audiological follow-up. All children were implanted by the senior author (VC). All patients underwent pure-tone audiometry (PTA) and speech perception testing in quiet (mono/disyllabic word lists) of the non-implanted ear at least 5 years after implantation. 

Exclusion criteria were children who were unable to complete PTA testing due to cognitive disabilities or had developmental delay; diagnosed with auditory neuropathy; whose parents did not consent to CI or ABI even after diagnosis (*n* = 6).

Electrophysiological Investigation:

Evoked potentials were recorded either with the Amplaid MK12 or Medelec Synergy N-EP, Viasys system. The initial tests in the protocol included impedance audiometry and ABR to clicks. ABR outcomes determined the continuation of the investigation with RW-ECoG. In the presence of a normal ABR threshold, i.e., better than 30 dB nHL, no further investigation was performed. When the ABR thresholds indicated moderate, severe, or profound hearing loss, children underwent RW-ECoG. For RW-ECoG, a myringotomy was performed under otomicroscopy from the malleus umbo to the posterior annulus of the tympanic membrane to visualize the round window niche. Any effusion was evacuated. A cotton-wick recording electrode was then placed on the round window niche under direct view (Figure 1). The electrode consisted of a string of 7 silver wires insulated with Teflon and soldered at the distal end to a cotton wick of about 1.5 mm diameter. A small pledget of wet Surgicel was placed over the electrode to ensure stability and improve electrode impedance. Contralateral masking was used during ABR and RW-ECoG. 

Children with ABR and ECoG thresholds between 30 and 75 dB nHL were referred to an audiologist for a hearing aid trial. If RW-ECoG confirmed severe or profound hearing loss, i.e., a RW-ECoG threshold worse than 75 dB nHL, RW-EABR was also performed to differentiate a cochlear from a retrocochlear site of hearing loss. The same cotton-wick electrode previously used for recording was then used to perform RW-EABR in order to test the viability of the cochlear nerve fibers by means of contralateral EABR recording (Figure 2). Upon the completion of testing, the myringotomy was closed by approximating the incision margins and placing a tympanic membrane resorbable patch when deemed necessary by the surgeon.

Auditory Brainstem Responses (ABR). For ABR testing, needle electrodes were placed subcutaneously with the standard vertex to pre-tragus montage. Electrode impedance was smaller than 3 kΩ. Electrodes were connected to an evoked potential system and responses were band-pass filtered between 100 and 3000 Hz. Clicks of 100 μs duration, produced by a 100 μs electrical square wave, were presented via TDH-39 headphones at a rate of 21 per second with alternating polarity. Click signals were calibrated in dB nHL. At each presentation, the number of sweeps was set at 1024. Data were collected in time windows of 10–15 ms. The threshold was defined as the minimum stimulating level where an ABR response was both identifiable and repeatable. Starting at 120 dB nHL, clicks were presented twice for each sound intensity level in order to ensure reproducible responses. The level of stimulation was decreased in 20 dB nHL steps until no response was present. Thereafter, ABR was repeated using 10 dB steps and then 5 dB steps until the threshold was established. 

Round Window Electrocochleography (RW-ECoG). RW-ECoG was performed using the cotton-wick electrode over the round window niche, placed as described above (Figure 1). The electrode was connected to the positive input of the preamplifier and referenced to a needle electrode placed at the ipsilateral pre-tragus area. The same filter settings and recording parameters as for ABR recording were used. Stimuli were presented with alternated polarity to cancel the microphonic potentials. At each presentation level, 128 to 256 sweeps were averaged. Acoustic monophasic clicks of 100 μs duration were presented with the same protocol as used for ABR recording. Cochlear microphonics (CM) recordings were obtained with positive and negative polarity clicks. Action potential (AP) responses were evoked with clicks of alternating polarity. Short 4 ms tone bursts or logon of different frequencies were then presented to selectively stimulate regions of the basilar membrane, and the AP was recorded. Four frequencies were used to assess the cochlear responses: 500, 1000, 2000, and 4000 Hz. The stimulus was selected to be brief enough to evoke a synchronized response, but of sufficient duration to provide frequency specificity. Stimuli had the following characteristics: 500 Hz—rise and fall of 1 cycle with no plateau; 1000 Hz—rise and fall of 1 cycle and a plateau of 2 cycles; 2000 Hz—rise and fall of 2 cycles and a plateau of 4 cycles; 4000 Hz— rise and fall of 4 cycles and a plateau of 8 cycles. The thresholds for clicks and tone burst stimuli were obtained in a similar manner to that described for ABR testing above.

Round window electrical auditory brainstem responses (RW-EABR). After being used as a recording electrode for RW-ECoG, the same cotton-wick electrode placed on the round window niche was used to deliver electrical stimuli to test the viability of the cochlear nerve fibers. To minimize stimulus artifacts, biphasic rectangular pulses were delivered by a constant current stimulator to the cotton-wick electrode. The stimulating electrode was referenced to a sub-dermal needle electrode placed in the ipsilateral pre-tragus area. The electrical stimulus was a biphasic pulse of 100 μs per phase and its intensity ranged from 1 to 1.5 mA. EABR were recorded using a contralateral electrode montage with the positive electrode inserted at the vertex, the negative electrode at the contralateral pre-tragus area, and the ground at the forehead. Each recorded waveform consisted of 1024 samples of EEG activity over a 10 ms time base, and the results were band-pass filtered between 1 and 2500 Hz. The EABR threshold was defined as the minimum current level required to evoke an identifiable wave V. 

The best threshold identified with ABR or ECoG testing was used to define the preoperative electrophysiological threshold for the frequency of the stimulus used. Before undergoing implantation, children underwent a trial with hearing aids, vibrotactile devices, or both with no improvement.

Statistical analysis was performed using the paired T-test and Mann–Whitney test as appropriate (we compared the implanted ear with the non-implanted ear). Statistical significance was set at *p* < 0.05. 

## 3. Results

### 3.1. Population

A total number of 790 subjects underwent the diagnostic protocol, including 401 males and 389 females. The median age at the time of diagnosis was 13.4 ± 9.6 months.

Out of the 790 tested children, 214 (27%) showed bilateral severe-to-profound hearing loss. Among these children, 13 (6.1%) also had a diagnosis of auditory neuropathy (absent ABR associated with normal CM) [24,25,26]. One hundred and fifty-eight children (20%) received a CI and fifty (6%) received an ABI. Six subjects did not perform surgery due to their parents’ refusal of the proposed treatment.

### 3.2. Outcomes of the Electrophysiological Investigation

Electrophysiological thresholds obtained with ABR and RW-ECoG tests are presented in Table 1. 

In 56 subjects (6.8%) with either ABR thresholds between 70 and 90 dB nHL or no response, RW-ECoG showed thresholds below 75 dB nHL (see Figure 2 for an example).

RW-EABR showed reliable responses in 161 subjects, of whom 158 received a CI (Figure 3). RW-EABR showed no response in 50 children. These children had various inner ear malformations diagnosed by MR and CT (e.g., cochlear nerve deficiency, cochlear malformation, or both), and received an ABI.

No short-term or long-term complications (e.g., persistent tympanic membrane perforation) due to either the testing protocol or general anesthesia were identified. Test protocol duration was assessed in a subset of recording sessions (163 subjects). Test protocol duration for ABR is related to the number of intensity levels tested and ranged from 4 min to 15 min (mean 9 ± 4 min). The RW-ECoG portion, including myringotomy and placement of the electrode, lasted between 8 min and 20 min (mean 14 ± 5 min). RW-EABR testing lasted between 4 and 11 min (mean 8 ± 3 min). Total electrophysiological protocol duration ranged from 28 to 65 min (mean 41 ± 13 min).

### 3.3. Pure Tone Audiometry

To further assess the efficacy of our protocol, hearing thresholds obtained at initial diagnosis by electrophysiological methods were compared to PTA at least 5 years after the implantation. Twenty-one infants diagnosed with bilateral profound sensorineural hearing loss before 6 months of age and fitted with a unilateral cochlear implant before 12 months underwent pure-tone audiometry (PTA) of the non-implanted ear at least 5 years after the implantation. Demographics and clinical data of these cases, including the etiology of hearing loss if known, are presented in Table 2.

None of the implanted children had functional hearing in the non-operated ear, defined as at least one threshold better than or equal to 75 dB nHL for 250, 500, or 1000 Hz. 

In 11 children (52.5%), PTA confirmed the previous diagnosis of profound hearing loss (no responses up to 120 dB nHL), with no differences between pre- and post-operative PTA in the non-operated ear.Eight children (38%) had hearing worse than 100 dB nHL at frequencies below 2000 Hz and two children (9.5%) showed a flat residual hearing worse than 100 dB nHL. No statistically significant differences in the paired T-test and Mann–Whitney test were identified between auditory thresholds of the non-implanted ear between the results at diagnosis (115.8 ± 4.9 dB nHL) and at the last follow-up (113 ± 6.2 dB nHL) among the 8 subjects (Figure 4; *p* > 0.05) with residual hearing below 2 kHz (0.5 and 1 kHz) (Table 3).Residual hearing was not detectable on the implanted side in any children. Speech discrimination scores at the last follow-up with the cochlear implant off were 0%.

The mean speech perception outcome at the last follow-up for the 21 children implanted (implant ON) was 85.2 ± 10.7. 

## 4. Discussion

As a result of the critical importance of auditory input early in life for speech and language development and the ability to intervene with hearing devices (hearing aids, CI, and ABI), many countries have adopted mandatory newborn hearing screening. Children deprived of auditory information in the first 3–5 years of life will have difficulty in oral communication even if their hearing is restored because their brain mechanisms for developing auditory pattern recognition will have passed the critical period of brain plasticity [27,28].

For many years, it was also thought that RW-ECoG was not reliable enough to predict hearing status and thus that the risks of the procedure (notably, general anesthesia and myringotomy) did not justify the limited benefit. However, as the present results show, RW-ECoG does have the reliability to diagnose severe and profound hearing loss in infants. This objective electrophysiological procedure, combined with RW-EABR, can be used to establish candidacy for CI or ABI in infants. RW-ECoG represents cochlear microphonics and summation potentials and is related to hair cell presence and health. RW-EABR, in contrast, is an evoked neural potential and indicates the presence and health of the auditory nerve. Recording these potentials from the round window provides a clear signal that has high reliability, selectivity, and specificity for hearing diagnosis. Children with damaged hair cells but an intact auditory nerve can benefit from a CI, while children with no response on either measure require an ABI because this indicates the absence of an auditory nerve, so a CI would be of no benefit. This lack of auditory nerve can be cross-checked with MRI results. Buchmann et al. (2011) [19] showed that children with auditory nerve deficiency as shown on MRI as well as poor EABR measures performed in the lower 10%-ile of CI outcomes, even after 10 years of use.

Furthermore, RW-ECoG demonstrated moderate hearing loss (<70 dB nHL) in 53/790 (6.8%) subjects with an ABR threshold between 70 and 90 dB nHL or no response. All these children were referred to the audiologist for hearing aid evaluation. Conversely, extratympanic ECoG seems less effective [29,30]. RW-ECoG has been demonstrated as reliable in monitoring hearing function during CI in children [22] and adults [31,32]. Recently, early implantation has also been associated with an improved rate of residual hearing preservation [33,34]. Another possible use of RW-ECoG and RW-EABR, in combination with MRI, concerns the selection of the children suitable for an ABI, prior to CI use and subsequent poor outcomes, avoiding the loss of time for speech development [35].

The combination of the presented extensive electrophysiological and radiological (CT and MRI) protocols is helpful in improving the efficacy of diagnosis, despite being mildly invasive. The extra information that RW-ECoG provided can be fundamental in view of a better threshold definition as well as to rule out auditory neuropathy. In cases of cochlear nerve hypoplasia, RW-EABR could add important information regarding the candidacy between CI and ABI.

The audiological evaluation of infants with a comprehensive protocol seems to be highly effective in subjects with profound hearing loss who are candidates for cochlear implantation below 1 year of age and even before 6 months. Residual hearing was detected in half of the 21 subjects who underwent CI, but no subjects showed serviceable residual hearing for hearing aid fitting.

Our results agree with Vlastarakos et al., who underline the safety and efficacy of the diagnosis and, eventually, CI in infants [3,13]. Regarding ABR outcomes, multifrequency response seems reliable for defining CI candidates only if the result is “no responce” [29]. Aimoni et al. [20], recently described a series of 23 infants who demonstrated significant improvement in hearing after 6 months for term-born children and up to 11 months in severe prematurity. Among these subjects, only 10 demonstrated profound bilateral hearing loss at the first assessment, and only 1 among the 8 described in detail in the text performed transtympanic ECoG under general anesthesia. In our opinion, these results are comparable with the outcomes of the present study in light of the fact that RW-ECoG seems to be the most sensitive test to define hearing levels in children due to the nearer field recording and the chance to improve middle ear effusion with suction.

The major limitations of the present study are the small number of children below 1 year of age implanted and that the non-implanted ear has been considered as the control group since they had substantially the same hearing threshold. Finally, as in all studies concerning cochlear implants, the heterogeneity of the population and multiple covariates must be taken into consideration.

In our experience, RW-ECoG and RW-EABR are at present the most reliable electrophysiologic techniques for establishing hearing thresholds and the site of lesion in infants with severe to profound hearing loss, furnishing useful indications for candidacy to CI or ABI in a quick, inexpensive, cost-effective, and reproducible way. The entirety of the procedure lasts on average 40′, a duration that may be achieved thanks to general anesthesia. The important information obtained from RW-ECoG and RW-EABR can avoid placing an unnecessary CI as well as placing a cochlear implant in a “dead” cochlea unsuitable for electrical stimulation.

## 5. Conclusions

A comprehensive electrophysiological evaluation of infants is highly reliable in identifying children with profound hearing loss who may be candidates for CI or ABI. Severe-to-profound loss was detected in half of the examined cohort, and some had minor residual hearing, but none showed residual hearing sufficient for hearing aid use. We propose RW-ECoG as a reliable method for diagnosing and defining hearing loss and for determining whether a hearing aid, CI, or ABI is the most appropriate management approach.

## Data Availability

Not applicable.

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
