# Peer review of "A Retrospective Evaluation to Assess Reliability of Electrophysiological Methods for Diagnosis of Hearing Loss in Infants"

_brainsci, 2022, doi:10.3390/brainsci12070950_

Round 1

Reviewer 1 Report

This is an interesting study that explores the relevant issue of reliability of electrophysiological procedures for diagnosing hearing loss in early infancy.  Background is clearly and exaustively presented, methods quite sound, results well presented and described. I've only some minor concern: language needs some revision, information about hearing recovered at implanted site could be useful, the meaning of the sentence at page 8:  "Residual hearing was not detectable on the implanted side in any children.
Speech discrimination scores at the last follow up without cochlear implant was  0%." is not clear, likely here authors are speaking about the preimplant state, however a further clarification would be better.

Reviewer 2 Report

Congratulations on putting together an interesting manuscript assessing an important topic of improving diagnostics for improved early interventions in infants. I had the following comments and some detailed comments (presented at the end).

If the final treatment outcome can be done reliably with just imaging why use an invasive procedure for diagnosis. it is not clear from the manuscript how much using these electrophysiological procedures improves our ability to decide candidature for CIs/ABIs over an above just using imaging. What would have been the protocol if these measurements were not made?

There are some issues with the presentation of information (see detailed comments) and organization of content that make the manuscript hard to read. Improving this might vastly improve the accessibility of the information presented.

I am just curious, how many of these 21(or total number of implantees) children showed improved outcomes after implantation?

I look forward to reading the revised manuscript.

Detailed Comments.

Abstract

P1. L15. Check case at beginning.

 Can you please reword the background to reflect that this is a retrospective study. I would recommend that you say that the effectiveness of the electrophysiological measures as diagnostic tools was assessed in this study.

P1. L19. Check case at beginning.

P1. L23-29. It is not clear how these results support your stated conclusions. It would make it easier to the reader if the

P2. Introduction. It would help if even a cursory introduction to RW ECoG and RW-EABR are provided to help readers understand why these tests were used.

P2. L73. Recommend changing to “underwent genetic, paedriatic, and neurophysiological evaluations”. It is not clear what paedriatic and neurophysiological evaluations are in this context.

P3. L 79-82. Just to clarify, you identified 214 children with bilateral profound sensorineural hearing loss according to the results in the abstract portion of the manuscript. Here you only talk about the 21 CI implantees. It’s not clear why you do that. It is not clear why this text is presented here. Please consider including text to say that this group was used to test efficacy of protocol through audiometry later.

P3. L86. I would suggest revising this to “Children who were unable to complete PTA testing due to cognitive disabilities or had developmental delay; diagnosed auditory neuropathy; whose parents did not consent to CI or ABI even after diagnosis (n=6) were excluded from the analysis.”

Figure 2. The text in all the figures is close to unreadable. Please include text to support the inclusion of this figure.

P6. L198. Response present/absent numbers here do not match those in Table 1.  

P7. L217. In the abstract and everywhere else, you say that these tests were done 5 years postop. Here you say 5 years of age.

P8. L233. Consider using a more specific term instead of “auditory thresholds”.  What thresholds are you referring to here? Please state clearly what was being compared here.  How did you convert PTA thresholds to dB nHL (scale used in Figure 4)?  

Why does Table 3 only have values for 500 and 1 kHz?

Table 3. Should the last row be threshold for 1k at last follow up?

P9. L264-268. This information should be presented in the introduction rather than here as the reason these tests were chosen.

P9. L293. “only if the result is “no response” … “
